# The Ghrelin Analog GHRP-6, Delivered Through Aquafeeds, Modulates the Endocrine and Immune Responses of *Sparus aurata* Following IFA Treatment

**DOI:** 10.3390/biology14080941

**Published:** 2025-07-25

**Authors:** Leandro Rodríguez-Viera, Anyell Caderno, Rebeca Martinez, Gonzalo Martinez-Rodríguez, Milagrosa Oliva, Erick Perera, Juan Miguel Mancera, Juan Antonio Martos-Sitcha

**Affiliations:** 1Department of Biology, Faculty of Marine and Environmental Sciences, Instituto Universitario de Investigación Marina (INMAR), University of Cadiz, Campus de Excelencia Internacional del Mar (CEIMAR), 11519 Puerto Real, Cadiz, Spain; anyell.caderno@uca.es (A.C.); milagrosa.oliva@uca.es (M.O.); juanmiguel.mancera@uca.es (J.M.M.); 2Aquatic Biotechnology Project, Center for Genetic Engineering and Biotechnology, Havana 10900, Cuba; rebeca.martinez@cigb.edu.cu; 3Andalusian Institute of Marine Sciences (ICMAN), Spanish National Research Council (CSIC), 11519 Puerto Real, Cadiz, Spain; gonzalo.martinez@csic.es (G.M.-R.); erick.perera@csic.es (E.P.)

**Keywords:** peptide-based immunomodulators, growth hormone secretagogue, humoral immune response, dietary supplement, gilthead sea bream

## Abstract

Aquaculture is growing rapidly, and improving the health and resilience of farmed fish is essential for sustainable production. This study tested whether a small synthetic molecule, GHRP-6, could be used as a feed additive to support the immune system of gilthead sea bream, an important farmed fish in the Mediterranean. After several weeks of feeding fish with a diet enriched with GHRP-6, the animals were exposed to a mild immune challenge. The fish that received the GHRP-6 diet showed signs of better immune readiness, with increased levels of immune-related molecules in the blood and specific genes activated in key tissues. These fish also had lower stress hormone levels compared to the control group, suggesting a protective effect. Importantly, no damage was observed in their internal organs. Together, the results suggest that GHRP-6 may help fish respond better to immune challenges while keeping stress under control. This could make fish farming more efficient and reduce the need for other treatments. Further research is needed to understand how this compound works and whether it can help fish resist infections in the long term. This study shows that smart nutrition can be a powerful tool in modern aquaculture.

## 1. Introduction

The aquaculture sector has experienced significant growth in recent decades. In fact, in 2022, for the first time, aquaculture surpassed capture fisheries in aquatic animal production, with 94.4 million tonnes, representing 51% of the world total and recording 57% of production for human consumption [1]. This accelerated sector growth was accompanied by the subsequent increase in feeding and nutritional issues for sustaining this activity, mainly related to using high-quality, safe, and environmentally friendly feed ingredients [1]. Although more feeds are formulated based on digestible nutrients for some species, such as Atlantic salmon, the improvement of growth rate and feed efficiency remains the main goal for most species.

The quality of the formulation depends on several factors or sets of knowledge, including a precise understanding of the nutrient requirements of the specific species, the nutritional composition of the ingredient used and its digestibility and bioavailability, but also on the processes for producing the aquafeeds [2]. The rapid development and the intensification of aquaculture, sometimes associated with hapless management practices, among other factors, brought pressures on the cultivated aquatic organisms, compromising their immunological defenses against various diseases and even the appearance of new diseases for marine organisms [3,4]. As a result, these diseases inflict significant limitations on animal growth, eligibility, and the sustainability of the aquaculture industry. To solve this problem, generic antibiotics started to be introduced into animal feed quite a few years ago. However, many issues came with this practice, from the emergence of resistant pathogens (antimicrobial resistance) and transmission to the human food chain to its impact on marine ecosystems [4].

To try to solve these problems, the employment of additives in aquafeeds has proven to be a suitable option to ensure the optimal use of the feed and improve different productive indicators in farmed fish [2,4]. Feed additives are defined (briefly) as those non-nutritive ingredients or components that are included in diet formulations, which have the capacity to influence the physicochemical properties of the diet, modifying the performance of the animals [5]. Nowadays, it is imperative to achieve the production of functional feed with an optimized nutritional composition and functional additives that allow better growth rates, immunity, and general health status of the animals, as well as stress resistance; these aspects allow us to promote a more sustainable aquaculture while ensuring the quality and safety of the products [2].

In the gilthead seabream (*Sparus aurata*), the use of additives such as short- or medium-chain fatty acids [6,7] or nutraceutical compounds from algae and bacteria [8,9], among others, has proved to be a suitable option to improve different productive indicators. Besides, the employ of synthetic compounds that stimulate feed intake, feeding efficiency, stress resilience, physiological pathways related to growth, metabolism, or immune enhancement may open new avenues to increasing the competence of this productive activity [10,11,12,13]. Nevertheless, few studies are available on farmed fish in this regard.

Growth hormone (GH) secretagogues (GHSs) or GH-releasing peptides (GHRPs) are a family of synthetic, non-natural peptides [14] recognized by the receptors of GH. The action of several synthetic GHRPs on GH secretion has been studied in different livestock [15], including fish [16,17]. For example, the synthetic peptide A233 has been shown to affect growth, antioxidant defense, and immune system function in tilapia fish [18,19]. Also, this decapeptide presents a potential antiviral effect in the gilthead sea bream and zebrafish (in vivo model) as well as in RTG-2 (a fibroblast cell line derived from gonad tissue of rainbow trout) [17]. Another strong GHRP is the Growth Hormone-Releasing Peptide 6 (GHRP-6), a six-amino-acid synthetic peptide [His-(D-Trp)-Ala-Trp-(D-Phe)-Lys-NH2] and considered a ghrelin analog, first described by Bowers et al. [20]. Few studies have been performed in fish regarding the effects of this synthetic peptide. In freshwater species, like goldfish, GHRP-6 mimics the orexigenic action of ghrelin [21]. In juvenile tilapia, this peptide increased pituitary GH secretion both in vitro and in vivo [18,22,23], enhanced hepatic insulin-like growth factor I (IGF-I) expression after intraperitoneal administration, enhanced antimicrobial peptide transcription, and stimulated growth rate when administered in the pharyngeal cavity of juvenile tilapia [23,24,25,26]. In addition, body weight and non-specific immunity enhancements have also been observed in tilapia larvae when they are subjected to immersion baths with this peptide [18,23]. While, in European seabass (*Dicentrarchus labrax*), recent studies demonstrated its antiviral effects against red-spotted nervous necrosis virus, both in cell culture and in vivo, highlighting its broader immunophysiological relevance in fish [27]. Also, in rainbow trout, a related synthetic peptide secretagogue (GHRP-2) stimulates feed intake and increases plasma GH, IGF-I, and IGF binding protein levels after intraperitoneal injections [28]. Apparently, GHRPs have a wide variety of actions in teleosts. However, our understanding of central and peripheral actions is not fully elucidated, and most available information comes from studies performed on freshwater species.

In a previous study, we evaluated the effect of dietary administration of GHRP-6 at two different levels, for 97 days, on specimens of *S. aurata*. Both experimental diets increased feed efficiency and growth performance, where the lower inclusion of GHRP-6 results in better aerobic metabolism. In comparison, the higher inclusion significantly enhanced plasma GH levels in agreement with the GH secretagogue effects of ghrelin [29]. In the present study, the effects of dietary administration of GHRP-6 peptide on the endocrine and immune responses of *S. aurata* following Incomplete Freund’s adjuvant (IFA) treatment were assessed. The findings provide new insights into how peptide-based nutritional interventions can influence both mucosal and systemic immune responses in teleost fish.

## 2. Materials and Methods

### 2.1. Animal Maintenance and Diets

Juveniles of gilthead sea bream (*S. aurata*) were acquired from a commercial source (PREDOMAR, Carboneras, Almería, Spain), randomly distributed in nine 400 L tanks, and acclimatized for ten days with seawater in controlled conditions of salinity (36 ppt) and temperature (19 °C) and under natural photoperiod at our latitude (36°31′45″ N, 6°11′31″ W) in the indoor experimental facilities at the *Servicios Centrales de Investigacion en Cultivos Marinos* at the University of Cadiz (SCI-CM, CASEM, UCA, Puerto Real, Cádiz, Spain; Spanish Operational Code REGA ES11028000312). Water quality parameters, including temperature, dissolved oxygen, and nitrogen compounds, were monitored regularly and remained stable throughout the experimental period. Fish were kept and handled following the guidelines for experimental procedures in animal research from the Ethics and Animal Welfare Committee of the University of Cádiz, according to the Spanish (RD53/2013) and European Union (2010/63/UE) legislation. The Ethical Committee from the Autonomous Andalusian Government approved the experiments (Junta de Andalucía reference number 04/04/2019/056).

The experimental diet was prepared from a commercial feed for gilthead sea bream (INICIO Plus 805, BioMar Iberia S.A., Palencia, Spain), which had a pellet size of 0.3 mm and the following composition: crude protein 50.0%, crude fat 18.0%, digestible carbohydrates 16.1%, crude fiber 2.4%, ash 8.0%, and phosphorus 1.1%. This base feed was supplemented with the synthetic GHRP-6 peptide at a concentration of 500 μg GHRP-6/kg of feed. Diet control was prepared only with the same excipients used to add the peptide to the experimental diet, excluding the peptide. GHRP-6 (His-(D-Trp)-Ala-Trp-(D-Phe)-Lys-NH2, MW = 872.44 Da, >99%) was acquired from Sigma-Aldrich, St. Louis, MO, USA. The feed was prepared as described previously in Rodriguez-Viera et al. [29], following the methodology developed by Adelmann et al. [30] with some modifications. In brief, the peptide reconstituted in phosphate-buffered saline was added to a mixture of aluminum hydroxide (10%) and polyethylene glycol 1000 (90%), which had been heated to 37 °C in a water bath. This suspension was then used to enrich pulverized commercial diet at final doses of 0 and 500 μg of GHRP-6 per kg of feed. Subsequently, pellets were formed using cold extrusion and dried at 22 °C.

### 2.2. Experimental Design and Sampling Procedure

A total of 180 juvenile fish (average initial weight: 20.6 ± 0.5 g; length: 10.55 ± 0.08 cm) were randomly allocated into nine 400 L tanks (30 fish per tank; 90 individuals per dietary group). Fish were maintained under controlled conditions for a period of 97 days. Experimental diets were offered to apparent satiation twice daily, during the early morning and late afternoon, to ensure maximum feed intake while minimizing waste. Each feed was labeled with a distinct color code without indicating its composition, allowing for a blind-feeding strategy that minimized potential bias in feed administration and fish handling, as described in Rodriguez-Viera et al. [29]. Fish were group-weighed and measured every three weeks. Feed intake was monitored weekly for each tank, enabling the calculation of feed efficiency (FE), expressed as the ratio between total weight gain (g) and total feed consumed (g) per tank during the experimental period. No mortality was observed throughout the trial.

At the end of the feeding period and with a body mass of 54.9 ± 4.69 g and body length of 14.6 ± 0.18 cm, 24 overnight-fasted fish were randomly selected from each dietary group (8 individuals per tank) and sedated with 2-phenoxyethanol (0.3 mL/L of seawater). Then, half of the selected fish (*n* = 12) were intraperitoneally injected with 100 µL of Incomplete Freund’s Adjuvant (IFA) to induce an immune response, while the other half (*n* = 12) received 100 µL of sterile saline solution as controls (Sal). Fish were maintained under the same experimental conditions and feeding regimen described above for three days. After 72 h post-injection, fish were deeply anesthetized with 1 mL of 2-phenoxyethanol/L of seawater, and blood and tissue samples were taken for further analysis (12 fish were per treatment, *n* = 48). Blood was drawn from caudal vessels with heparinized syringes, centrifuged at 4000× *g* for 15 min at 4 °C, and plasma samples were snap-frozen in liquid nitrogen and stored at −80 °C until biochemical and hormone analysis. Before tissue collection, fish were euthanized by cervical section. Samples of spleen (S), head kidney (HK), anterior intestine (AI), and posterior intestine (PI) for gene expression analyses were immediately placed in RNAlater at 4 °C for 24 h and then stored at −20 °C until total RNA extraction. Spleen, AI, and PI samples were also taken and fixed in phosphate-buffered saline at pH 7.2 containing formalin (10%) for histomorphological analysis.

### 2.3. Plasma Analyses

Plasma total protein content was quantified using the BCA Protein Assay Kit (PIERCE, Thermo Fisher Scientific, Waltham, MA, USA), with bovine serum albumin (BSA) serving as the standard. Levels of cholesterol, glucose, triglycerides, and lactate in plasma were assessed using commercial kits supplied by SPINREACT (Girona, Spain), adapted for use in 96-well microplate format. Plasma cortisol levels were measured with a commercial Cortisol Enzyme Immunoassay Kit from ARBORASSAYS (NCAL International Standard Kit, DETECTX, K003) and following the manufacturer’s protocols. Plasma osmolality was measured with a vapor pressure micro-osmometer (Fiske^®^ Micro-Osmometer, model 210, Norwood, MA, USA) and expressed as mOsm L^−1^.

Total immunoglobulin (Ig) in plasma was performed according to Panigrahi et al. [31], slightly modified. Briefly, a plasma sample was mixed with an equal volume of 12% solution of polyethylene glycol (PEG, 10,000 MW, Sigma) and incubated for 2 h at 4 °C. Then, samples were centrifuged at 5000× *g* for 15 min at 4 °C. The supernatant was taken. The protein content was measured (as described above). Differences among the protein values of the untreated and polyethylene glycol-treated samples correspond to the total Ig content and were expressed as mg mL^−1^.

All measurements were carried out using a Bio-Tek PowerWave 340 microplate spectrophotometer, with data processing performed through KCjunior Data Analysis Software (www.biotek.com, accessed on 1 June 2025) (Bio-Tek Instruments, Winooski, VT, USA). For spectrophotometric assays (e.g., glucose, lactate, triglycerides, and cholesterol), absorbance readings were performed at specific wavelengths recommended by each assay kit. The readings were performed at 340 nm for glucose, 562 nm for total proteins, 450 nm for cortisol, and 505 nm for the other metabolites. All samples were analyzed in duplicate, and only those with a coefficient of variation (CV%) lower than 10% between replicates were accepted.

### 2.4. Gene Expression

A total of 12 fish were sampled per treatment (*n* = 48), and portions of HK, AI, PI, and S were homogenized by a Precellys^®^ Evolution Homogenizer (Bertin Instruments, Montigny-le-Bretonneux, France). Total RNA was isolated by means of the NucleoSpin^®^ RNA kit (Macherey-Nagel, Madrid, Spain). RNA quality was assessed with a 2100 Bioanalyzer (Agilent Technologies, Santa Clara, CA, USA), and quantification was performed using the Qubit™ RNA Broad Range Assay Kit on a Qubit^®^ 2.0 Fluorimeter (Thermo Fisher Scientific^®^). Only samples exhibiting RNA Integrity Numbers above 8.0 were selected for gene expression analysis. For cDNA synthesis, 500 ng of RNA from each sample were reverse transcribed in a 20 μL reaction using the qScript^™^ cDNA Synthesis kit (Quanta BioSciences^™^, Beverly, MA, USA) on a Mastercycler^®^ proS (Eppendorf AG, Hamburrg, Germany). The resulting cDNAs were diluted 1:10 in 10 mM Tris 0.1 mM EDTA (pH 8.0), yielding a final concentration of 2.5 ng/µL. Before conducting gene expression analyses, a calibrator sample was used to determine the optimal quantitative PCR (qPCR) conditions for each gene. For this, a pool of cDNAs from all samples was used, and six 1/10 serial dilutions (from 10 ng to 100 fg) were made to obtain calibration curves for each pair of primers. The R^2^ values were greater than 0.980, and efficiencies were between 90.0% and 110.0% in all cases. Control reactions using RNase-free water (NTC) and RNA (NRT) were included in the analysis to ensure there were no primer dimers and no genomic DNA contamination.

Using Hard-Shell^®^, white well/white shell, low-profile, thin-wall, 96-well, skirted PCR plates covered with Microseal^®^ B Adhesive Seals (BioRad, Hercules, CA, USA), each reaction was performed in duplicate with 0.5 μL (final concentration 200 nM) of each pair of primers, 5 μL PerfeCTa™ SYBR^®^ Green FastMix™ (Quanta Bio, Beverly, MA, USA), and 4 μL of cDNA (10 ng). The qPCR reactions were performed in a CFX Connect™ Real-Time PCR System (Bio-Rad Laboratories, Hercules, CA, USA) as follows: An initial denaturation and polymerase activation at 95 °C for 10 min, followed by 40 cycles of denaturation for 15 s at 95 °C, annealing and extension at 60 °C for 30 s, and finishing with a melting curve from 60 °C to 95 °C, increasing by 0.5 °C every 5 s. Melting curves were used to ensure that only a single PCR product was amplified and to verify the absence of primer-dimer artifacts. Relative gene expressions were analyzed by CFX Manager™ software (https://www.bio-rad.com/, accessed on 1 June 2025) (Bio-Rad), using the ΔΔC_T_ method [32] and corrected for efficiencies [33]. To normalize the results, actin beta 1 (*actb*) and eukaryotic elongation factor 1 alpha (*eef1a*) were chosen as internal reference genes. These two reference genes were selected due to their low variability in the current experiment across each tissue (CV < 0.25 and M < 0.5), meeting the requirements established by the GenNorm Target Stability Value. The genes analyzed are shown in Table 1, and the primer information is in Appendix A.

### 2.5. Histomorphological and Histochemical Analysis

Samples of the anterior and posterior segments of the intestine and spleen were collected from six fish in each experimental group (C-Sal, C-IFA, D-Sal, D-IFA) for the analysis (*n* = 24). For the intestines, the surrounding adipose and connective tissues were carefully removed from the samples before fixation. The histological tissue samples were fixed in phosphate-buffered saline at pH 7.2 containing formalin (10%). Then, the preparations were washed in running tap water, dehydrated in alcohol, cleared in xylene, and embedded in paraffin wax. Sections (6 μm) were cut and mounted on gelatinized slides using a rotary microtome. Sections were rehydrated in distilled water and stained with hematoxylin/eosin (H&E) [34]. Periodic Acid-Schiff (PAS) + Alcian Blue (AB) staining technique was used for specific histochemical reactions to visualize carbohydrates and glycoproteins [35].

### 2.6. Morphometric Analyses

The highest quality cross-section from each animal was selected for morphological measurements (six in each experimental group). We divided the surface mucosal length by the perimeter submucosa length (magnification 10×) to create a mucosal/submucosa surface ratio (MSR) from four measurements (diametrically opposed) of the cross section (4 per fish, 20 measurements for the experimental group). Goblet cell (GCs/mm^2^) number was calculated by the average of all neutral goblet cells observed (magnification 10×) from four areas (diametrically opposed) of the cross section (4 per fish, 20 measurements for experimental group). In the spleen, melanomacrophage centers (MMCs/mm^2^) were calculated in sections where the maximum diameter of the spleen was reached (magnification 2×) from a cross section (2 per fish, 10 measurements for experimental group). For morphometric analysis, micrographs were taken using a light microscope (Eclipse Ci-L, Nikon Corporation, Tokyo, Japan) and a Jenoptik ProgRes CT5 camera (Jenoptik AG, Jena, Germany). The biometric index was determined using Fiji software (https://fiji.sc/, accessed on 1 June 2025) (Image J2, National Institutes for Health, Bethesda, MD, USA).

### 2.7. Statistical Analyses

All data were tested for normality and homogeneity of variances using the Kolmogorov-Smirnov and Levene’s tests, respectively, with a significance threshold of *p* ≤ 0.05. Statistical differences between treatments were analyzed by two-way ANOVA, followed by Tukey’s post hoc test to compare group means (*p* ≤ 0.05). Results are presented as mean ± standard error of the mean (SEM). All statistical analyses and figure generation were carried out using GraphPad Prism version 8.0 (GraphPad Software, Inc., San Diego, CA, USA).

Additionally, a Linear Discriminant Analysis (LDA) was conducted to identify patterns associated with immune response. A total of 36 key variables were selected, encompassing gene expressions in the head kidney, spleen, anterior intestine, and posterior intestine, as well as plasma Ig levels in *S. aurata*. The MASS (https://cran.r-project.org/web/packages/MASS/index.html) (accessed on 1 June 2025) and caret (https://cran.r-project.org/web/packages/caret/index.html) (accessed on 1 June 2025) libraries were used for data analysis, and the ggplot2 (https://cran.r-project.org/web/packages/ggplot2/index.html) (accessed on 1 June 2025) package was employed for visualization. Data analysis was performed using RStudio software version 4.4.0 (24 April 2024).

## 3. Results

### 3.1. Plasma Biochemistry

No significant differences were observed between experimental diets regarding circulating glucose and total protein levels (Table 2). However, following the IFA challenge, plasma lactate concentrations were significantly reduced (*p*-value = 0.019) in fish fed the GHRP-6-supplemented diet (1.43 ± 0.11 mM) compared to the control group (2.00 ± 0.11 mM). Notably, lactate values in the GHRP-6 group were more similar to those recorded in fish injected with saline solution, regardless of diet (C-Sal and D-Sal). On the contrary, animals fed with a control diet significantly decreased triglyceride values when challenged with IFA (1.31 ± 0.41 mM) with respect to animals that were fed with a supplemented diet (*p*-value = 0.014), even when challenged with IFA (Table 2). Nevertheless, no differences were found in circulating cholesterol levels in animals challenged with the IFA, regardless of treatment. However, in saline-injected groups, fish fed the peptide-supplemented diet showed higher plasma cholesterol levels compared to those on the control diet. Interestingly, within the peptide-fed groups, cholesterol levels decreased in fish challenged with IFA compared to their saline-injected counterparts (Table 2). Plasma osmolality significantly decreased in fish fed the GHRP-6-supplemented diet and challenged with IFA (D-IFA), compared to the D-Sal group, indicating a significant diet-treatment interaction (*p* = 0.019). No differences were observed between the C-Sal and C-IFA groups (Table 2).

### 3.2. Immunoglobulin and Cortisol in Plasma

Immunoglobulin levels in plasma varied significantly between treatments (*p* = 0.014). As expected, fish injected with IFA had enhanced circulating immunoglobulin levels compared to their saline-injected counterparts. Notably, fish that received both a GHRP-6-supplemented diet and IFA treatment (D-IFA group) exhibited the highest levels of plasma immunoglobulins (4.67 ± 0.65 mg ml^−1^), which were significantly higher than those observed in the control diet group injected with saline or IFA. The group fed the GHRP-6 diet without IFA injection (D-Sal) also showed moderately elevated immunoglobulin levels (Table 2). Fish fed a control diet and challenged with IFA exhibited high circulating cortisol values. However, the D-Sal group presented higher values with respect to the C-Sal group, while in the group receiving the GHRP-6 diet and submitted to IFA treatment, these values decreased significantly (Table 2).

### 3.3. Gene Expression

#### 3.3.1. Anterior Intestine

The transcriptional response in the different tissues studied showed differential modulation of several immune-related genes in response to a dietary GHRP-6-supplemented diet and immune challenge with IFA (Figure 1). In the anterior intestine, the cytokines, *il10*, *il15*, and *il34* were significantly upregulated in fish fed a GHRP-6-supplemented diet and when challenged with IFA (*p* < 0.05). *il10* enhanced its expression levels in animals that consumed a GHRP-6-supplemented diet. However, *il15* and *il34* showed similar expression patterns of increasing in both C-IFA and D-Sal groups, while D-IFA decreased expression values (Figure 1). The *il8* showed no significant differences across treatments, although a mild increase was observed in the D-Sal group (Table 3). The expression of *mx1* was significantly upregulated in the D-IFA group with respect to all others (*p* < 0.05), while the *mx2* gene, although it was also slightly higher in D-IFA, showed that there were no differences in expression across groups (*p* > 0.05). Interestingly, *lgals1* was significantly downregulated in animals that ingested a dietary GHRP-6-supplemented diet (Sal and IFA) compared to controls (*p* < 0.05) (Figure 1). whereas *lgals8* expression did not vary significantly (Table 3).

Among the chemokine receptors, *ccr9* expression was similar in the D-IFA group relative to both treatments fed with the control diet and surprisingly significantly upregulated in the D-Sal group (*p* < 0.05), while remaining almost stable (*p* < 0.05) (Figure 1), whereas *ccr3* expression shown significantly differs between the IFA-treated groups (Table 3). No significant differences were found for *lgals8*, *muc2*, *ighm*, and *klf4* (Table 3).

#### 3.3.2. Posterior Intestine

In the posterior intestine, expression of most immune-related genes remained stable across treatments (*mx2*, *ccr9*, *klf4*, *lgals1*, *lgals8*, *il8*, *il15*, and *il34*; *p* > 0.05) (Table 3), although some significant differences were observed (Figure 2). Expression of the *mx1* gene was significantly downregulated in the D-Sal group compared to the C-Sal group (*p* < 0.05), while the C-Sal and D-IFA groups showed the highest values. Regarding mucosal markers, *muc2* expression increased in the C-IFA group with respect to other groups (*p* < 0.05). Moreover, the expression of muc3b revealed significant differences between animals fed dietary GHRP-6 and the control group. Finally, *ccr3* expression was not affected by diet, while IFA treatment decreased its values in both groups (C-IFA and D-IFA).

#### 3.3.3. Spleen and Head Kidney

In the spleen, several genes associated with immune regulation and antiviral response were significantly modulated by dietary GHRP-6 and immune stimulation (Figure 3). The pro-inflammatory cytokines *il8* and *il10* showed a similar pattern of change, significantly enhancing their expression in the D-IFA group compared to all others (*p* < 0.05). Moderate upregulation was also observed in the C-IFA group, while groups treated with saline solution exhibited significantly lower levels. Moreover, the expression of il34 revealed significant differences between animals challenged with IFA, with significantly less expression in the D-IFA group (Figure 3). No differences were found in the expression of *casp1* between the experimental groups or treatments (*p* > 0.05) (Table 3). Expression of *lgals1* was stable across treatments, with a slight, non-significant increase in D-IFA with respect to C-IFA and D-Sal, while C-Sal showed the lowest values and was significantly different from D-IFA (*p* < 0.05). In general, the two antiviral markers studied in the spleen (*mx1* and *mx2*) showed a similar pattern, reaching higher expression levels in C-Sal and D-IFA groups. Meanwhile, the C-IFA group had the lowest expression for both markers, especially *mx2* (*p* < 0.05). On the other hand, the expression of *ighm* was significantly higher in fish that received a GHRP-6 supplementation diet and were challenged with IFA (*p* < 0.05). In contrast, the other groups presented moderate (C-Sal) or lower (C-IFA and D-Sal) expression values for *ighm*, without differences between them.

Finally, among all the genes tested, we only detected two cytokines in the head kidney, *il10* and *il15*. No differences were found for *il10* expression levels (*p* > 0.05) (Table 3), while for *il15*, animals treated with GHRP-6 peptide and challenged with IFA showed the highest expression levels (*p* < 0.05) (Figure 3).

### 3.4. Multivariate Analysis

Figure 4 shows the results of the linear discriminant analysis (LDA). The first, second, and third discriminant functions explained 61.06%, 25.76%, and 13.18% of the total variability among the groups, respectively. The D-IFA and C-Sal groups were closer to each other (positive values) along the first linear discriminant function (LDA1), while the C-IFA and D-Sal groups were more closely associated with one another (negative values). LDA1 was primarily influenced by the expression of *il8* and *mx1* in the spleen, *mx1* in the posterior intestine, and *il10* in the head kidney, all of which had high positive contribution coefficients (Table 4). Conversely, *il15* in the anterior intestine and *mx2* and *ccr3* expression in the posterior intestine exhibited high negative coefficients (Table 4). The second discriminant function (LDA2) clearly separated the groups based on diet, regardless of the treatment. Individuals fed the control diet were distributed toward positive values mostly along LDA2, while those fed the D diet were located toward the negative side. However, the D-IFA group appeared closer to those fed the control diet (Figure 4). In LDA2, the variables that contributed most to group separation were the expression levels of *ccr3* and *muc3b* (with negative coefficients) and *mx2* (with a positive coefficient), all measured in the posterior intestine. In contrast, LDA3 was mainly influenced by *lgals1* expression in the anterior intestine and *mx1* in both the posterior intestine and spleen, which were associated with negative coefficients (Table 4).

### 3.5. Histomorphological and Histochemical Analysis of the Intestine and Spleen

Fish from the different experimental groups presented normal histological architecture in both the intestine and spleen (Appendix A). The ratio of the mucosal/submucosal surface, as well as the number of goblet cells in the intestine and the count of melanomacrophage centers in the spleen (Table 5), do not show differences between experimental groups (*p* > 0.05).

## 4. Discussion

Dietary administration of GHRP-6 has been shown to enhance growth performance in the gilthead sea bream [29], reinforcing its value as a functional feed additive in this species. Thus, the present study goes one step further and evaluates the immunomodulatory effects of dietary GHRP-6 supplementation in *S. aurata* by examining gene expression in immune-relevant tissues, plasma biochemical markers, and systemic immune indicators after an immunological challenge (with IFA treatment). The findings provide new insights into how peptide-based nutritional interventions can influence both mucosal and systemic immune responses in teleost fish, reinforcing the known relationship between nutrition and immunology.

### 4.1. Plasma Biochemistry

Plasma biochemical analysis revealed treatment-specific metabolic responses. Thus, focusing both on the significant differences and the trends observed, the increase in lactate, together with triglyceride values decreasing in fish from the C-IFA group, supports the existence of a metabolic burst after intraperitoneal immunostimulation, providing the energy required by these substrates to cope with immunological challenge. Interestingly, cholesterol level enhancement observed in all groups with respect to the C-Sal group possibly reflected a stimulated lipoprotein synthesis associated with immune cell membrane remodeling or steroid biosynthesis produced by IFA injection or by GHRP-6 itself, as previously observed during active immune responses in fish [36,37].

Cortisol levels were significantly elevated in C-IFA and D-Sal groups but notably reduced in D-IFA fish, indicating that dietary GHRP-6 may mitigate the stress response induced by adjuvant injection. This pattern aligns with previous findings in gilthead sea bream and other teleosts where functional diets or peptide-based additives attenuated stress-induced cortisol release [38,39,40].

Interestingly, fish from the D-Sal group, fed with the GHRP-6-supplemented diet but not challenged with IFA, exhibited elevated plasma cortisol levels compared to the C-Sal group. This suggests that GHRP-6 may modulate baseline endocrine activity, potentially through activation of the hypothalamic-pituitary–interrenal (HPI) axis. As a known ghrelin analog and GH secretagogue, GHRP-6 interacts with endocrine pathways that influence not only growth but also metabolism and stress-related processes [17]. One plausible explanation is that GHRP-6 induces a state of physiological “priming”, a preparatory activation of neuroendocrine and immune responses in anticipation of potential challenges. This moderate activation may serve adaptive purposes, such as increasing metabolic readiness or mobilizing immune resources. Such priming could be reflected in moderately elevated cortisol levels in the absence of an acute inflammatory trigger. Similar mechanisms have been described in other teleosts, where basal cortisol elevation is associated with enhanced immunocompetence or energy mobilization under mild stimulation [41,42]. Summarizing, our findings indicate that dietary GHRP-6 can play a dual role in modulating the stress response in *S. aurata*. It appears to reduce cortisol levels during inflammatory challenges while simultaneously inducing a basal activation of the HPI axis under non-stressed conditions. This context-dependent modulation highlights the potential of GHRP-6 as a functional dietary additive but also underscores the need for further studies exploring its endocrine effects under different physiological scenarios.

Moreover, plasma osmolality is a key indicator of the hydromineral balance and cellular homeostasis in fish [43], particularly under stress conditions such as immune stimulation. In the present study, a significant interaction between diet and treatment was observed for plasma osmolality, with the lowest values recorded in the D-IFA group. This reduction contrasts with the elevated osmolality found in the D-Sal group (Table 2), suggesting that the combination of dietary GHRP-6 and immune challenge modulates osmoregulatory processes in *S. aurata*. The decrease in plasma osmolarity observed in the D-IFA group may reflect improved diet-induced homeostatic regulation of GHRP-6 under conditions of immunological stress. This response appears to be mediated by reduced cortisol and lactate mobilization. It could represent a protective effect of the peptide against the disruption of hydromineral and metabolic balance induced by the IFA adjuvant [44].

It is well known that IgM is the most prevalent immunoglobulin in serum and is responsible for systemic immunity [45]. In our study, plasma Ig concentrations were markedly increased in the D-IFA group, consistent with enhanced spleen *ighm* expression, and suggested that GHRP-6 supplementation stimulates the humoral system of the immune system in synergy with inflammatory stimuli. A similar pattern was observed in *Oreochromis* spp., where GHRP-6 treatment followed by *Pseudomonas aeruginosa* challenge led to a significant upregulation of sIgM expression in the spleen, indicating a localized humoral immune activation in response to the peptide stimulus [26]. This condition may be particularly advantageous in aquaculture, where enhancing the fish’s antibody-mediated defenses can reduce the incidence or severity of disease outbreaks. Thus, dietary GHRP-6 supplementation could represent a feasible strategy to support immune readiness in *S. aurata* under farming conditions, ultimately contributing to healthier stocks and reducing the need for pharmacological interventions.

### 4.2. Gene Expression

Gene expression analysis revealed tissue-specific modulation in response to GHRP-6 inclusion in aquafeeds. From an immunological perspective, the anterior intestine plays a central role in initiating mucosal immune responses due to its early and direct exposure to dietary antigens and potential pathogens, displaying the most dynamic transcriptional response, particularly under inflammatory stimulation. It is well established that this segment harbors a higher density of immunocompetent cells, including macrophages, dendritic-like cells, and intraepithelial lymphocytes, which contribute to antigen sampling and immune regulation [46,47].

In our study, GHRP-6 supplementation led to elevated expression of *il10*, a key anti-inflammatory cytokine that mediates mucosal tolerance, alongside *il15* and *il34*, which are involved in lymphocyte survival and macrophage activation, respectively. These findings are in line with previous studies in *S. aurata* and other teleosts, where oral administration of immunostimulants triggered similar cytokine responses [44,48,49]. In our case, an increase in mRNA expression of these key genes in GHRP-6-fed fish could suggest an enhancement of both regulatory and Th1-type immune responses in gut mucosa, consistent with previous findings in teleosts linking these cytokines to immune homeostasis and pathogen defense [50,51].

Interestingly, expression of the antiviral gene *mx1* was significantly increased only in fish fed a GHRP-6 diet and challenged with IFA, indicating a synergistic effect between dietary peptide and immune stimulation. This agrees with previous studies showing that *mx1* is highly responsive to dietary immunostimulants and viral mimics in *S. aurata* [27,51]. Furthermore, *mx* gene upregulation has also been reported following administration of synthetic growth hormone secretagogues, such as peptide A233, reinforcing the link between this peptide class and antiviral immune activation [17]. Although *mx2* did not vary significantly in our study, its trend toward elevation in the D-IFA group may reflect a delayed or secondary phase of antiviral activation, consistent with observations from in vivo teleost models [17,52].

The chemokine receptor *ccr9*, related to lymphocyte trafficking in gut-associated lymphoid tissues, was upregulated in the D-Sal group, which could suggest enhanced immune surveillance even in the absence of overt inflammation. However, the downregulation of *lgals1* in GHRP-6-fed fish is intriguing, as galectin-1 is commonly associated with anti-inflammatory and tissue repair processes [53,54]. This finding may reflect a shift in immune balance or modulation of regulatory pathways by GHRP-6. Moreover, the absence of significant differences in *muc2*, *ighm*, or *klf4* expression in the anterior intestine, with the slight increases in *muc2* and *ighm* detected in GHRP-6 groups, suggests a trend toward mucosal fortification. All together, these data support the notion that GHRP-6 primes the anterior intestine for a more effective immune response without compromising barrier integrity or inducing excessive inflammation, even when an immunostimulatory process is not busted.

In contrast, the response of the posterior intestine to dietary intervention and immunostimulation is more modest. Downregulation of *muc2* expression in GHRP-6 groups may reflect modulation of mucosal barrier components, although the biological significance of this reduction remains unclear and warrants further investigation. The expression stability of *lgals1*, *lgals8*, and *ccr9* in this region suggests region-specific immune modulation along the gut axis, a phenomenon already described in teleosts, including *S. aurata* [50].

In the spleen, *il8* and *il10* were strongly upregulated in the D-IFA group, indicating a balanced pro- and anti-inflammatory response under combined peptide and immune challenge. Notably, *ighm* expression increased significantly in these fish, suggesting a potential protective role of GHRP-6 in supporting humoral immunity. Similarly, in the head kidney, elevated *il15* expression in the D-IFA group points to systemic immune activation driven by dietary peptide. These findings echo recent studies showing that immunonutritional strategies can enhance cytokine responses and immunoglobulin synthesis in sea bream and other teleosts [49,55,56].

The biochemical and gene expression results further support an immune-relevant role for GHRP-6. Lower lactate and cortisol levels in the D-IFA group compared to controls may reflect reduced stress and metabolic demand, suggesting a buffering effect of the peptide during immune activation. Moreover, the observed increase in plasma IgM levels in GHRP-6-fed fish aligns with the transcriptional data from the spleen, reinforcing the hypothesis that this peptide supports both innate and adaptive immune responses.

### 4.3. Histomorphological and Histochemical Analysis

In the present study, no significant changes were observed in intestinal or splenic histomorphology of *S. aurata* following dietary supplementation with GHRP-6 or after intraperitoneal injection with IFA. Goblet cell number and the mucosa-to-submucosa ratio remained consistent across all groups, indicating that neither peptide nor adjuvant induced detectable structural alterations in these tissues.

Although the potential impact of GHRP-6 on gut morphology in fish has not been previously characterized, earlier work by Rodríguez-Viera et al. [29] demonstrated that dietary GHRP-6 significantly increased growth performance and feed efficiency in *S. aurata* without altering key organosomatic indices such as hepatosomatic and mesenteric indices (HSI and MSI), as well as intestinal length index (ILI). Interestingly, in carnivorous species such as gilthead sea bream, intestinal length has been shown to display plasticity in response to dietary changes [8,57], which could contribute to improved nutrient absorption. However, our current results do not suggest that GHRP-6 induces notable morphological adaptations at the intestinal level over the time frame tested. It remains to be determined whether longer exposures or different doses of GHRP-6 could promote changes in mucosal architecture or epithelial cell populations, such as increased absorptive surface or modulation of goblet cell numbers, features previously associated with feed efficiency gains [6]. However, while longer exposures or higher doses of GHRP-6 may enhance its biological effects, the economic feasibility of such approaches under intensive aquaculture conditions must be considered. Although the current dietary inclusion level is low (1%) and the peptide is synthetically produced, which could support its cost-effectiveness, scaling up dosages could still affect overall feed costs. Therefore, future research should not only assess the physiological efficacy of different dosages and exposure durations but also evaluate cost-effectiveness to ensure scalability and adoption in commercial aquafeeds.

Regarding the IFA adjuvant, the lack of inflammatory signs in intestinal histology is also notable. Classic markers of intestinal inflammation in fish include mucosal fold shortening and goblet cell hyperplasia [58,59], while systemic inflammatory responses to IFA have been reported in salmonids and cod, including adhesive peritonitis and pyloric caeca inflammation [60,61]. However, the IFA dose used in our study (100 µL per 100 g fish) was considerably lower than that employed in those previous reports (250–500 µL), which may explain the absence of detectable lesions. Thus, while we cannot conclusively attribute anti-inflammatory protection to GHRP-6 based on histology alone, our findings suggest that neither peptide nor IFA triggered overt tissue disruption under the conditions tested.

In the spleen, melanomacrophage centers (MMCs) are often used as histological indicators of chronic immune stimulation [62,63]. However, in our study, they did not increase in number in any group, including IFA-injected fish, suggesting again the use of a lower IFA dosage or the transient nature of the inflammatory stimulus. Overall, the lack of structural damage or excessive immune activation is consistent with the biochemical and molecular findings of a balanced immune modulation, reinforcing the potential of GHRP-6 as a safe functional additive in aquafeeds. It is worth noting that the absence of histological signs of inflammation following IFA injection may not be entirely unexpected, given the well-documented differences in innate immune recognition pathways between teleosts and mammals. In particular, fish show a marked resistance to the endotoxic effects of lipopolysaccharide (LPS), a commonly used proinflammatory stimulus in mammals [45]. This is partly attributed to the structural divergence or even absence of the canonical TLR4 receptor in many fish species. Even in species such as zebrafish, which possess TLR4 orthologs, LPS does not activate the NF-κB pathway as in mammals but rather suppresses it [64]. Moreover, studies that have employed LPS in fish models often require doses of several orders of magnitude higher than in mammals to observe any measurable immune activation [45]. These differences may also extend to other oil-based adjuvants like IFA, whose immunostimulatory mechanisms rely in part on the recruitment and activation of innate sensors. Therefore, the lack of clear intestinal or splenic inflammatory responses observed in our study may reflect not only the relatively low IFA dose used and/or the transient nature of the inflammatory stimulus but also fundamental differences in immune system sensitivity and activation thresholds in teleosts, which highlights the need for further research to better characterize these mechanisms across species.

### 4.4. Multivariate Analysis

Taken together, the transcriptomic profiles, along with the biochemical and hormonal analyses, reinforce the proposed role of GHRP-6 as an immunomodulatory peptide capable of buffering the stress and metabolic demands associated with inflammatory challenges. Notably, fish fed with GHRP-6 maintained more stable levels of plasma cortisol, lactate, and triglycerides following IFA injection, supporting a mitigating effect on systemic stress. Although histomorphological analyzes showed no overt structural alterations in intestinal or splenic tissues, multivariate analysis (LDA) revealed a distinct immunological signature among experimental groups, primarily driven by differences in gene expression. The clear separation observed in the LDA, particularly influenced by the expression of mx1, il10, il15, and ccr3, underscores the relevance of these genes as biomarkers of dietary immune modulation. Interestingly, the spatial proximity between D-IFA and C-Sal groups in the LDA suggests that GHRP-6 supplementation may attenuate the inflammatory transcriptional profile normally triggered by IFA, potentially contributing to a more balanced immune response under challenging conditions.

## 5. Conclusions

This study demonstrates that dietary supplementation with GHRP-6 peptide modulates key components of the immune system in *S. aurata*, enhancing both local and systemic responses. The coordinated upregulation of anti-inflammatory, antiviral, and humoral markers across intestinal and lymphoid tissues, along with increased plasma Ig levels and reduced cortisol under immune challenge, highlights the dual immunostimulatory and stress-buffering properties of GHRP-6. However, the exact cellular mechanisms underlying these effects remain unclear. The results, along with previous findings that showed improved growth performance and efficiency when this peptide was used as a dietary supplement, support the feasibility of incorporating GHRP-6 as an additive in diets adapted for *S. aurata*. Thus, future research should focus on long-term effects, pathogen resistance, and possible synergistic formulations with other immunonutrients to optimize its application, even exploring its putative long-term effects on immune memory. Understanding these mechanisms will be key to optimizing the use of GHRP-6 and related peptides in sustainable aquafeed strategies.

## Figures and Tables

**Figure 1 biology-14-00941-f001:**
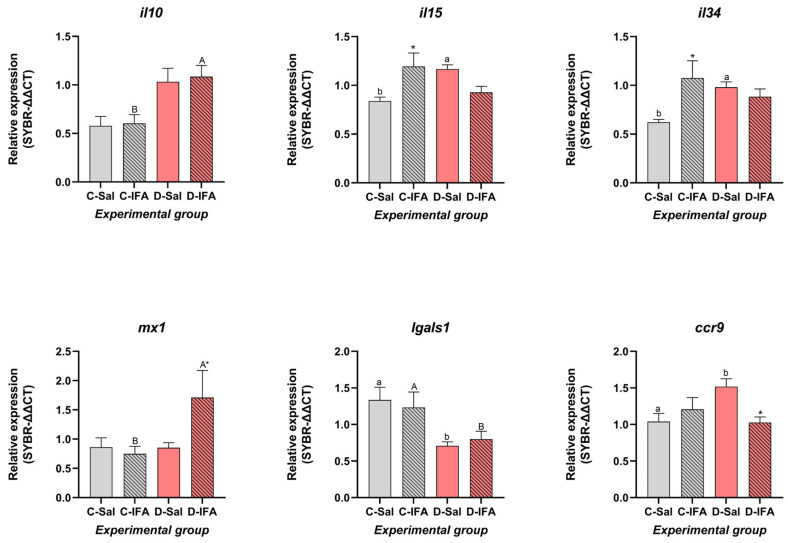
Gene expression level in the anterior intestine of juvenile gilthead sea breams (*Sparus aurata*) fed with a commercial (C) diet or supplemented with 500 µg GHRP-6/kg aquafeed (D) for 97 days, injected with 100 µL/g fish of saline solution (Sal) or Incomplete Freund’s adjuvants (IFA), and sampled after 72 h post-injection. Each value is the mean ± SEM of 12 fish. Different capital letters indicate significant differences among groups challenged with the IFA (C-IFA and D-IFA), lowercase letters indicate differences among groups that received saline (C-Sal and D-Sal). An asterisk denotes significant differences within the same group (Control or GHRP-6), regardless of the treatment challenge (two-way ANOVA, Tukey test, *p* ≤ 0.05). Statistical values are provided in Appendix A.

**Figure 2 biology-14-00941-f002:**
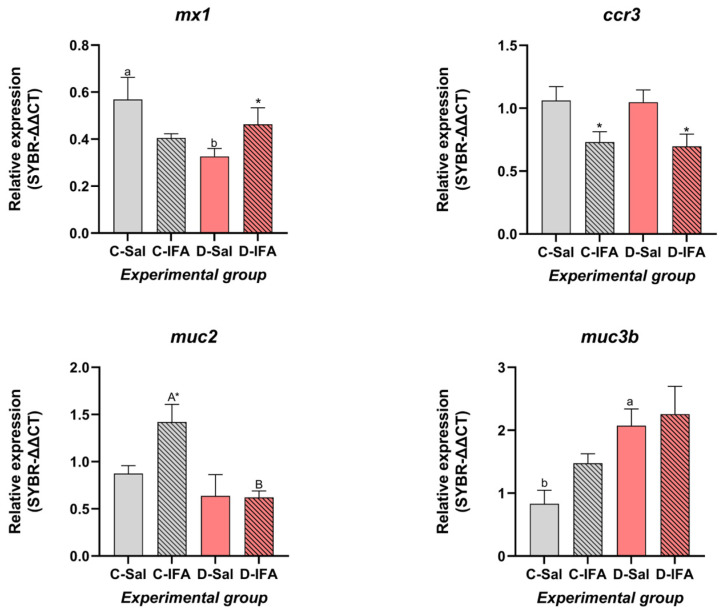
Gene expression level in the posterior intestine of juvenile gilthead sea breams (*Sparus aurata*) fed with a commercial (C) diet or supplemented with 500 µg GHRP-6/kg aquafeed (D) for 97 days, injected with 100 µL/g fish of saline solution (Sal) or Incomplete Freund’s adjuvants (IFA), and sampled after 72 h post-injection. Each value is the mean ± SEM of 12 fish. Different capital letters indicate significant differences among groups challenged with the IFA (C-IFA and D-IFA), and lowercase letters indicate differences among groups that received saline (C-Sal and D-Sal). An asterisk denotes significant differences within the same group (Control or GHRP-6), regardless of the treatment challenge (two-way ANOVA, Tukey test, *p* ≤ 0.05). Statistical values are provided in Appendix A.

**Figure 3 biology-14-00941-f003:**
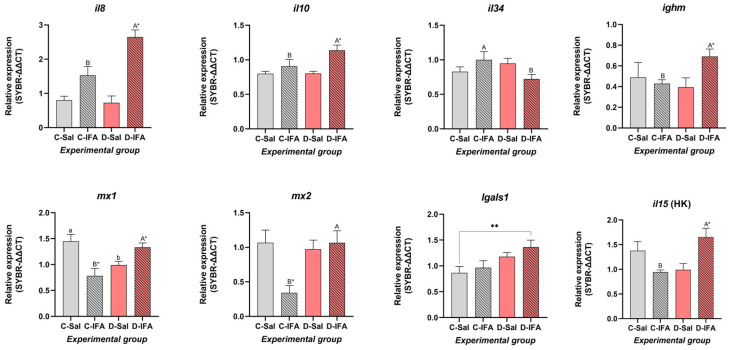
Gene expression level in the spleen and head kidney (HK) of juvenile gilthead sea breams (*Sparus aurata*) fed with a commercial (C) diet or supplemented with 500 µg GHRP-6/kg aquafeed (D) for 97 days, injected with 100 µL/g fish of saline solution (Sal) or Incomplete Freund’s adjuvants (IFA), and sampled after 72 h post-injection. Each value is the mean ± SEM of 12 fish. Different capital letters indicate significant differences among groups challenged with the IFA (C-IFA y D-IFA), and lowercase letters indicate differences among groups that received saline (C-Sal y D-Sal). Asterisks denote significant differences within the same group (Control or GHRP-6), regardless of the treatment challenge (two-way ANOVA, Tukey test, * *p* < 0.05, ** *p* < 0.01). Statistical values are provided in Appendix A.

**Figure 4 biology-14-00941-f004:**
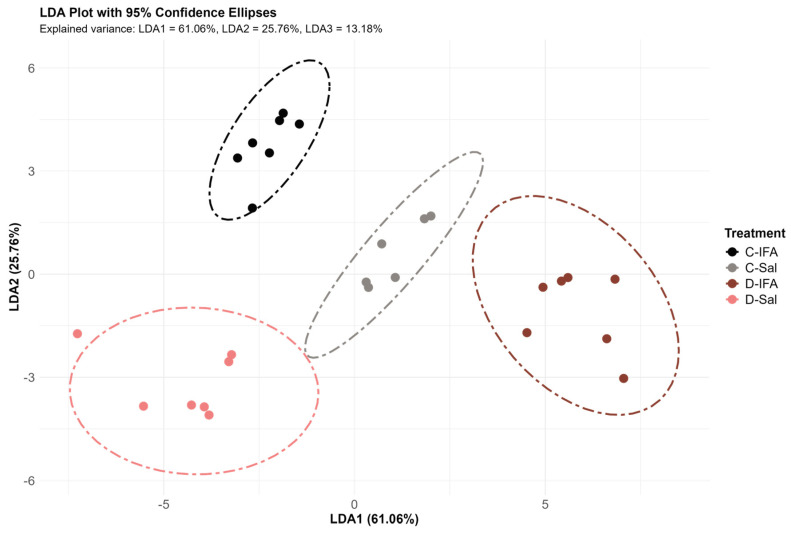
Linear discriminant analysis (LDA) describing patterns related to the immune response, based on relevant variables including gene expression levels in the head kidney, spleen, anterior intestine, and posterior intestine, as well as plasma immunoglobulin levels in juvenile gilthead sea breams (*Sparus aurata*) fed with a commercial (C) diet or supplemented with 500 µg GHRP-6/kg aquafeed (D) for 97 days, injected with 100 µL/g fish of saline solution (Sal) or Incomplete Freund’s adjuvants (IFA), and sampled after 72 h post-injection.

**Table 1 biology-14-00941-t001:** Immune system genes analyzed in the gilthead sea bream *S. aurata*.

Gene	Name	Accession Number
Reference genes		
*actb1*	Actin beta 1	XM_030406939.1
*eef1a*	Eukaryotic elongation factor 1 alpha	AF184170.1
Immunological response/inflammatory status		
*il8*	Interleukin-8	JX976619.1
*il10*	Interleukin-10	XM_030418889.1
*il15*	Interleukin-15	JX976625.1
*il34*	Interleukin-34	XM_030427145.1
*casp1*	Caspase 1	XM_030438153.1
*lgals1*	Galectin-1	KF862003.1
*lgals8*	Galectin-8	KF862004.1
*ccr3*	c-c chemokine receptor type 3	XM_030401704.1
*ccr9*	c-c chemokine receptor type 9	XM_030397691.1
*mx1*	Interferon-induced GTP-binding protein Mx1	FJ490556.1
*mx2*	Interferon-induced GTP-binding protein Mx2	FJ490555.1
*ighm*	Immunoglobulin m	XM_030408618.1
Mucus production and goblet cell differentiation		
*muc2*	Mucin 2	JQ277710.1
*muc3b*	Mucin 3b	XM_030418634.1
*klf4*	Krueppel-like factor 4	XM_030435936.1

**Table 2 biology-14-00941-t002:** Plasma biochemistry of juvenile gilthead sea breams (*Sparus aurata*) fed with a commercial (C) diet or supplemented with 500 µg GHRP-6/kg aquafeed (D) for 97 days, injected with 100 µL/g fish of saline solution (Sal) or Incomplete Freund’s adjuvants (IFA), and sampled after 72 h post-injection. Each value is the mean ± SEM of 12 fish. Different capital letters indicate significant differences among groups challenged with the IFA (C-IFA and D-IFA) lowercase letters indicate differences among groups that received saline (C-Sal and D-Sal). An asterisk denotes significant differences within the same group (Control or GHRP-6), regardless of the treatment challenge (two-way ANOVA, Tukey test, *p* ≤ 0.05).

Metabolites	Experimental Groups						
	C-Sal	C-IFA	D-Sal	D-IFA	*p*-Diet	*p*-IFA	*p*-Interaction
Glucose (mM)	3.57 ± 0.16	4.01 ± 0.18	3.72 ± 0.09	3.72 ± 0.13	0.613	0.140	0.135
Protein (mg ml^−1^)	34.29 ± 1.15	33.58 ± 1.76	38.86 ± 1.03	33.74 ± 1.92	0.148	0.078	0.175
Lactate (mM)	1.44 ± 0.17	2.00 ± 0.12 ^A^*	1.52 ± 0.08	1.43 ± 0.11 ^B^	0.079	0.098	0.027
Triglyceride (mM)	2.55 ± 0.19	1.31 ± 0.41 ^B^*	2.67 ± 0.25	2.64 ± 0.36 ^A^	0.030	0.054	0.068
Cholesterol (mM)	6.02 ± 0.35 ^b^	7.37 ± 0.67	8.54 ± 0.42 ^a^	7.25 ± 0.55 *	0.006	0.304	0.008
Hormones and antibodies							
Cortisol (ng ml^−1^)	6.79 ± 0.78 ^a^	34.05 ± 6.41 ^A^*	42.0 ± 4.48 ^b^	17.27 ± 1.94 ^B^*	≥0.001	0.095	<0.0001
Immunoglobulin (mg ml^−1^)	1.64 ± 0.50	2.03 ± 0.41 ^B^	3.03 ± 0.72	4.67 ± 0.65 ^A^	0.004	0.1077	0.3042
Osmolality (mOsm/L^−1^)	289.6 ± 1.70	291.6 ± 5.41	298.76 ± 3.13	279.0 ± 4.74 *	0.687	0.051	0.019

**Table 3 biology-14-00941-t003:** Gene expression level of juvenile gilthead sea breams (*Sparus aurata*) fed with a commercial (C) diet or supplemented with 500 µg GHRP-6/kg aquafeed (D) for 97 days, injected with 100 µL/g fish of saline solution (Sal) or Incomplete Freund’s adjuvants (IFA), and sampled after 72 h post-injection. Each value is the mean ± SEM of 12 fish. An asterisk denotes significant differences within the same group (Control or GHRP-6), regardless of the treatment challenge (two-way ANOVA, Tukey test, *p* ≤ 0.05).

Genes Relative Expression (SYBR-ΔΔCT)	Experimental Groups						
	C-Sal	C-IFA	D-Sal	D-IFA	*p*-Diet	*p*-IFA	*p*-Interaction
Intestine anterior							
*il8*	0.88 ± 0.08	1.00 ± 0.13	1.26 ± 0.17	1.00 ± 0.19	0.192	0.663	0.204
*mx2*	0.55 ± 0.12	0.55 ± 0.10	0.72 ± 0.09	0.77 ± 0.22	0.159	0.868	0.851
*lgals8*	0.88 ± 0.06	0.84 ± 0.09	1.07 ± 0.09	0.88 ± 0.11	0.216	0.235	0.416
*ccr3*	0.85 ± 0.05	1.10 ± 0.15 *	0.82 ± 0.06	1.03 ± 0.13 *	0.650	0.031	0.844
*muc2*	0.81 ± 0.16	0.73 ± 0.09	1.19 ± 0.18	0.84 ± 0.09	0.077	0.123	0.311
*ighm*	1.00 ± 0.19	0.94 ± 0.21	1.28 ± 0.17	1.39 ± 0.17	0.194	0.609	1.000
*klf4*	0.88 ± 0.16	0.83 ± 0.11	0.55 ± 0.07	0.73 ± 0.13	0.105	0.604	0.356
Intestine posterior							
*mx2*	0.73 ± 0.15	0.91 ± 0.33	0.50 ± 0.06	0.83 ± 0.23	0.526	0.299	0.769
*ccr9*	1.02 ± 0.07	0.99 ± 0.09	1.10 ± 0.15	0.95 ± 0.11	0.882	0.419	0.575
*il8*	1.05 ± 0.11	1.11 ± 0.15	1.11 ± 0.14	1.05 ± 0.14	0.987	0.974	0.627
*il15*	1.10 ± 0.08	1.08 ± 0.12	1.12 ± 0.15	1.19 ± 0.12	0.587	0.809	0.730
*il34*	0.93 ± 0.06	0.84 ± 0.07	0.86 ± 0.13	0.94 ± 0.07	0.880	0.940	0.290
*klf4*	0.77 ± 0.13	0.89 ± 0.04	0.76 ± 0.15	0.74 ± 0.08	0.469	0.642	0.498
*lgals1*	0.79 ± 0.14	1.10 ± 0.10	1.14 ± 0.19	1.10 ± 0.15	0.230	0.382	0.234
*lgals8*	0.63 ± 0.14	1.09 ± 0.14	0.58 ± 0.20	0.70 ± 0.17	0.505	0.293	0.355
Spleen							
*casp1*	0.84 ± 0.14	0.84 ± 0.13	0.63 ± 0.08	0.68 ± 0.09	0.113	0.828	0.821
Head kidney							
*il10*	1.12 ± 0.14	1.25 ± 0.15	1.22 ± 0.12	1.63 ± 0.28	0.281	0.173	0.464

**Table 4 biology-14-00941-t004:** Contribution coefficients of each variable to the first three linear discriminant functions (LD1–LD3). Each value represents the loading (coefficient) of a variable on the respective linear discriminant axis obtained from LDA. Variables include gene expression levels in the head kidney (HK), spleen (S), anterior intestine (AI), posterior intestine (PI), and plasma immunoglobulin (Ig) levels in the gilthead sea breams *Sparus aurata*.

Variable	LD1	LD2	LD3
Anterior intestine			
*il8*	−0.53	0.01	−0.01
*il10*	0.21	−0.61	−0.66
*il15*	−1.40	0.36	0.69
*il6*	0.58	−0.07	−0.11
*mx1*	−0.20	−0.31	0.11
*mx2*	−0.25	−0.28	0.67
*lgals1*	0.10	−0.03	−0.98
*lgals8*	0.11	0.19	0.05
*ccr3*	0.52	0.17	0.31
*ccr9*	0.58	−0.20	−0.47
*muc2*	−0.06	−0.24	−0.20
*ighm*	−0.55	0.02	0.21
*klf4*	−0.18	0.18	0.44
Posterior intestine			
*il8*	−0.79	−0.24	0.20
*il15*	0.17	0.26	0.11
*il34*	−0.72	−0.42	−0.04
*mx1*	1.38	0.51	−0.94
*mx2*	−0.85	0.73	−0.24
*lgals1*	0.30	−0.17	−0.32
*lgals8*	0.59	0.13	−0.05
*ccr3*	−0.84	−1.01	0.24
*ccr9*	−0.21	0.61	0.29
*muc2*	−0.48	0.62	0.23
*muc3b*	0.18	−0.94	0.67
*klf4*	0.49	0.20	−0.01
Head kidney			
*il10*	1.01	−0.02	−0.30
*il15*	0.07	0.49	0.47
Spleen			
*il8*	1.74	0.34	1.16
*il10*	0.42	0.26	0.24
*il34*	−0.76	0.58	0.21
*mx1*	1.48	−0.34	−0.74
*mx2*	0.60	−0.42	−0.37
*casp1*	−0.30	−0.64	−0.52
*lgals1*	−0.38	−0.30	0.25
*ighm*	−0.07	0.15	0.45
Plasma			
Ig	0.07	−0.33	−0.63

**Table 5 biology-14-00941-t005:** Morphometric parameters in the intestine and spleen of the gilthead sea breams *S. aurata* under different experimental conditions. MSR (Mucosa-Submucosa Ratio); GC (Goblet Cells/mm^2^); MMC (Melanomacrophages centers/mm^2^). Significant differences analyzed (*p* ≤ 0.5).

Morphometric Parameters and Tissue	Experimental Groups			
	C-SAL	C-IFA	D-SAL	D-IFA
MSR-AI	7.36 ± 0.78	7.66 ± 0.50	6.69 ± 1.21	8.20 ± 0.80
MSR-PI	5.53 ± 0.57	5.59 ± 0.29	6.08 ± 0.67	5.53 ± 0.29
GCs-AI	834.80 ± 43.14	719.98 ± 39.78	885.45 ± 80.40	802.17 ± 47.30
GCs-PI	1539.97 ± 92.01	1405.24 ± 140.15	1235.43 ± 117.68	1436.18 ± 66.13
MMCs-S	4.53 ± 0.52	5.68 ± 1.36	6.21 ± 1.16	4.03 ± 0.27

## Data Availability

The original contributions presented in this study are included in the article and the Appendix A. Further inquiries can be directed to the corresponding author.

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
