# Peer review of "The Ghrelin Analog GHRP-6, Delivered Through Aquafeeds, Modulates the Endocrine and Immune Responses of Sparus aurata Following IFA Treatment"

_biology, 2025, doi:10.3390/biology14080941_

Round 1
Reviewer 1 Report
Comments and Suggestions for Authors
The manuscript falls within the scope of the journal Biology and presents important and innovative results and information regarding Sparus aurata. Overall, the article is well written, but it requires some additions and corrections that will improve the clarity and comprehension of the information.
Abstract: It is appropriate.
Keywords: It is suggested that the words “ghrelin” and “metabolism” be replaced with terms that are more specific to the research.
Introduction:
- Page 2, lines 41 to 71: The paragraph is too long. It is recommended to divide it into two or three shorter paragraphs.
Materials and Methods:
- Page 3, line 118: The total number of animals used in the experiment must be stated.
- Page 3, lines 118 to 123: The characteristics of the facilities where the fish were kept must be provided (e.g., tanks or containers? Volume in m³, acclimation equipment). Information on the water quality parameters (oxygen, temperature, nitrogen compounds, alkalinity, etc.) measured during the experiment must also be added. Additionally, the acclimation period for the fish in the facilities must be specified.
- Page 3, lines 129 and 130: The nutritional characteristics of the feed used (particle size, crude protein, fat content, etc.) should be described.
- Page 3, lines 134 to 140: The information presented is very similar to that in the article published by Rodriguez-Viera et al. [29]. To avoid plagiarism issues (in this case, self-plagiarism), it is recommended to rephrase and modify the content.
- Page 4, lines 142 to 145: The experimental design should be clearly stated (it is not possible to understand what the experimental unit refers to without this information), including the number of tanks per treatment and the number of fish per treatment/experimental diet. Referring to the study by Rodriguez-Viera et al. [29] limits access to essential experimental information (even if identical, the two studies have different objectives). Therefore, a detailed description of these aspects should be included in the text.
- Page 4, lines 142 to 145: The two feeding times should be mentioned. Since feeding was done ad libitum, it should be stated whether leftover feed was weighed (if any) to correct for daily feed intake. The method used to calculate feed consumption during the experiment should also be explained.
- Page 4, line 146: The meaning of “(n = 24)” is unclear. Does it refer to 24 fish per tank? 24 fish per treatment?
- Page 4, lines 149 to 151: What does “ii)” refer to? Were there no blood and tissue samples collected before the administration of Incomplete Freund’s Adjuvant (IFA)?
- Page 4, line 160: Delete “(see below)”.
- Page 6, line 217: If 12 fish per treatment/diet were used for plasma and gene expression analyses, how many were used for histomorphological, histochemical, and morphometric analyses? This should be clearly stated in the methodology. The use of the total number of fish per treatment/diet (90) must be clarified, and if some fish were not used, that should also be indicated.
Results:
- Page 4, lines 148 to 150: The methodologies for Shannon index, Chao1 index, and Simpson index analyses were not provided.
Discussion:
- Page 14, lines 428 to 452: This content does not belong in the Discussion section, as it is not discussing the results; rather, it reads more like an introduction. It is recommended to delete this paragraph and, if possible, use some of the information to enhance the Introduction.
- Page 14, lines 458 to 462: This statement appears speculative and is not supported by the results obtained. It is recommended to remove it.
- Page 15, line 467: To strengthen this hypothesis, it is requested that additional studies be analyzed (preferably from 2023 to 2025), beyond the one already cited (Sitjà-Bobadilla et al., 2005).
- Page 15, lines 498 to 502: The reasoning in this discussion is sound but must be supported by bibliographic references (preferably from 2023 to 2025).
- Page 16, lines 503 to 510: The statement is correct, but it is necessary to describe how this condition could support aurata farming in captivity.
- Page 17, lines 586 to 590: It is understood that longer exposures or higher doses might enhance the effect of GHRP-6; however, from an economic perspective, this might not be a viable option in fish nutrition under captive conditions. This issue warrants further discussion.
- Page 18, lines 619 to 622: It is essential to clarify that further research is required to improve the understanding of immune system sensitivity and activation thresholds. Some references could be cited here to support this concept.
- Page 18, lines 624 to 634: This part of the discussion could be improved.
Conclusion: It is appropriate.
Author Response
Reviewer 1
The manuscript falls within the scope of the journal Biology and presents important and innovative results and information regarding Sparus aurata. Overall, the article is well written, but it requires some additions and corrections that will improve the clarity and comprehension of the information.
Authors: We are very grateful for your suggestions and comments, and we have addressed each aspect suggested. We consider them to have greatly improved the manuscript.
Keywords: It is suggested that the words “ghrelin” and “metabolism” be replaced with terms that are more specific to the research.
Authors: The keywords were modified to read: peptide-based immunomodulators, growth hormone secretagogue, humoral immune response, dietary supplement, and gilthead sea bream.
Introduction:
- Page 2, lines 41 to 71: The paragraph is too long. It is recommended to divide it into two or three shorter paragraphs.
Authors: We agree, the paragraph is divided into three shorter ones.
Materials and Methods:
- Page 3, line 118: The total number of animals used in the experiment must be stated.
Authors: Corrected in the manuscript
- Page 3, lines 118 to 123: The characteristics of the facilities where the fish were kept must be provided (e.g., tanks or containers? Volume in m³, acclimation equipment). Information on the water quality parameters (oxygen, temperature, nitrogen compounds, alkalinity, etc.) measured during the experiment must also be added. Additionally, the acclimation period for the fish in the facilities must be specified.
Authors: All suggestions are clarified in the Materials and Methods section. See sub-index 2.1.
- Page 3, lines 129 and 130: The nutritional characteristics of the feed used (particle size, crude protein, fat content, etc.) should be described.
Authors: Information was added to the text.
- Page 3, lines 134 to 140: The information presented is very similar to that in the article published by Rodriguez-Viera et al. [29]. To avoid plagiarism issues (in this case, self-plagiarism), it is recommended to rephrase and modify the content.
Authors: Thank you for pointing this out. We acknowledge that the methodology presented in lines 134–140 shares similarities with our previously published work. We have carefully rephrased and reformulated the content in the revised version of the manuscript to avoid redundancy and ensure originality, while still accurately describing the experimental procedure. We hope this resolves the concern regarding potential self-plagiarism.
- Page 4, lines 142 to 145: The experimental design should be clearly stated (it is not possible to understand what the experimental unit refers to without this information), including the number of tanks per treatment and the number of fish per treatment/experimental diet. Referring to the study by Rodriguez-Viera et al. [29] limits access to essential experimental information (even if identical, the two studies have different objectives). Therefore, a detailed description of these aspects should be included in the text.
Authors: Thank you for your valuable comment. We fully agree with your observation, and in response, we have revised and rewritten the corresponding section to provide a detailed and self-contained description of the experimental design. This includes the number of tanks per treatment, the number of fish per tank, and the assignment of fish to each experimental diet. We understand the importance of presenting all essential methodological information within the current manuscript, regardless of similarities with previous studies. See in 2.2 section.
- Page 4, lines 142 to 145: The two feeding times should be mentioned. Since feeding was done ad libitum, it should be stated whether leftover feed was weighed (if any) to correct for daily feed intake. The method used to calculate feed consumption during the experiment should also be explained.
Authors: Thank you for your valuable comment. We agree with your suggestion and have clarified this information in the revised manuscript. The specific feeding times have been included, and details regarding feed intake monitoring and the calculation method for feed consumption have been clearly described in the text.
- Page 4, line 146: The meaning of “(n = 24)” is unclear. Does it refer to 24 fish per tank? 24 fish per treatment?
Authors: The notation “(n = 24)” refers to the total number of fish per treatment group. This clarification has been added to the manuscript to avoid any confusion.
- Page 4, lines 149 to 151: What does “ii)” refer to? Were there no blood and tissue samples collected before the administration of Incomplete Freund’s Adjuvant (IFA)?
Authors: The label “ii)” was removed/adjusted in the text to avoid any confusion.
- Page 4, line 160: Delete “(see below)”.
Authors: Deleted.
- Page 6, line 217: If 12 fish per treatment/diet were used for plasma and gene expression analyses, how many were used for histomorphological, histochemical, and morphometric analyses? This should be clearly stated in the methodology. The use of the total number of fish per treatment/diet (90) must be clarified, and if some fish were not used, that should also be indicated.
Authors: At the end of the feeding period, 24 fish were randomly selected from each dietary group (8 individuals per tank). Then, half of the selected fish (n = 12) were intraperitoneally injected with 100 µL of Incomplete Freund's Adjuvant (IFA) to induce an immune response, while the other half (n = 12) received 100 µL of sterile saline solution as controls (Sal). For a total of 48 fish, all 12 fish from each group were used for plasma and gene expression analyses, while only 6 samples per treatment were taken (n = 24) for histomorphological, histochemical, and morphometric analyses. The number of fish and samples in each analysis is described in the text.
Results:
Page 4, lines 148 to 150: The methodologies for Shannon index, Chao1 index, and Simpson index analyses were not provided.
Authors: I think this is a mistake, right? No Shannon index, Chao1 index, and Simpson index analyses were performing.
Discussion:
- Page 14, lines 428 to 452: This content does not belong in the Discussion section, as it is not discussing the results; rather, it reads more like an introduction. It is recommended to delete this paragraph and, if possible, use some of the information to enhance the Introduction.
Authors: We agree with your observation. The long introductory paragraph of the discussion was removed.
- Page 14, lines 458 to 462: This statement appears speculative and is not supported by the results obtained. It is recommended to remove it.
Authors: Removed
- Page 15, line 467: To strengthen this hypothesis, it is requested that additional studies be analyzed (preferably from 2023 to 2025), beyond the one already cited (Sitjà-Bobadilla et al., 2005).
Authors: More recent references on the topic are added, although not much has been said about it.
Page 15, lines 498 to 502: The reasoning in this discussion is sound but must be supported by bibliographic references (preferably from 2023 to 2025).
Authors: More recent references on the topic are added, although not much has been said about it.
- Page 16, lines 503 to 510: The statement is correct, but it is necessary to describe how this condition could support aurata farming in captivity.
Authors: We appreciate this valuable comment. In the revised manuscript, we have expanded the paragraph to clarify how the observed enhancement of the humoral response may benefit Sparus aurata aquaculture. Specifically, we now highlight that elevated circulating IgM levels and increased ighm expression in lymphoid tissues could contribute to improved disease resistance and vaccine responsiveness in cultured fish. These effects are particularly relevant under intensive farming conditions, where animals are frequently exposed to opportunistic pathogens and stressors. Therefore, the immunostimulatory effect of dietary GHRP-6 may provide a functional advantage in promoting fish health, reducing reliance on antibiotics, and supporting more sustainable farming practices.
- Page 17, lines 586 to 590: It is understood that longer exposures or higher doses might enhance the effect of GHRP-6; however, from an economic perspective, this might not be a viable option in fish nutrition under captive conditions. This issue warrants further discussion.
Authors: Thank you for this valuable observation. We agree that economic viability is a crucial aspect in assessing the practical applicability of functional feed additives. In response, we have added a short paragraph in the discussion acknowledging the trade-off between biological efficacy and production cost, and emphasizing the need for future studies to also consider the cost-benefit balance of GHRP-6 supplementation strategies in aquaculture.
- Page 18, lines 619 to 622: It is essential to clarify that further research is required to improve the understanding of immune system sensitivity and activation thresholds. Some references could be cited here to support this concept.
Authors: Thank you for your suggestion. While we acknowledge the importance of supporting this statement, we believe that—given the preliminary nature of our findings and the broad variability reported in teleost immune responses—a general call for further research is appropriate at this stage. However, we have revised the sentence slightly to clarify this point.
- Page 18, lines 624 to 634: This part of the discussion could be improved.
Authors: We appreciate the reviewer's comment and agree that this paragraph required clarification. We have revised the section to improve its clarity and coherence, emphasizing the connection between the transcriptomic, biochemical, hormonal, histological, and multivariate findings. Specifically, we restructured the paragraph to better reflect how GHRP-6 supplementation modulates the immune response, highlighting the role of relevant biomarkers such as mx1, il10, il15, and ccr3, and discussing the implications of the LDA results more explicitly.
Authors: Thank you for your thoughtful comments and valuable suggestions. We believe that your recommendations have significantly improved the overall quality and clarity of the manuscript.
Reviewer 2 Report
Comments and Suggestions for Authors
The present study evaluated the effect of GHRP-6 as dietary additive in Spaurus aurata on endocrine and immune responses following IFA treatment. The manuscript sounds well-designed and written. It presents the background, aims, M&M, and results in a well-structured manner. The topic addressed by the authors is relevant, mainly in the context of modern aquaculture, which faces an emerging demand for more sustainable diets, and alternative additives to enhance fish health and resilience, avoiding the use of antibiotics, for instance.
The results suggest that GHRP-6 may affect positively both endocrine and immune responses, as demonstrated by changes in plasma biochemistry and gene expression. Specifically, following IFA treatment, fish fed the GHRP-6 supplemented diet showed greater resistance to stress, as evidenced by the stability of the stress-related biomarkers. In addition, there was an upregulation of genes associated with anti-inflammatory and immune responses. These and previous results may contribute to the development of more sustainable nutritional alternatives in aquafeeds.
The manuscript is well-written overall, and the figures and tables are informative and appropriately presented. Therefore, I suggest a few minor revisions:
General concepts comments:
The introduction is well-structured; the reader can identify the problem underlying the objectives. The authors provide up-to-date and relevant literature supporting the use of GHRPs to improve growth, immune system, and defense responses in other teleosts. The objectives are well defined, and the rationale is well supported by the authors’ previous findings on feed efficiency and growth performance, supporting the new questions regarding its effect on endocrine and immune responses. The literature is current and consistent with the subject.
However, the authors should consider providing references for lines 58-59: “To solve this problem, generic antibiotics started to be introduced into animal feed quite a few years ago.”
In the Material and Methods section, it would be important to provide to the readers some brief information regarding the plasma analyses, for example, the limit of the accepted coefficient of variation (CV%) between replicates, the wavelength used for readings, the inter and intra- assay variation, validation tests for the species, if available. The authors should as well include the references for the doses of both GHRP-6 (500 ug/kg of feed) and IFA (100 uL/100 g fish) used in the present study. This is particularly important for IFA, given that the authors later discuss that it could be a weak dose for triggering a stronger effect.
The results and discussion are easy to follow, figures and tables are well presented. If possible, it would be valuable to include micrographs of each morphometric parameters in the intestine and spleen analyzed as a supplementary figure, to better illustrate the histological observations. Although the first paragraph of the discussion is very informative, I suggest briefly summarizing the main findings overall to position the reader and emphasize the study’s significance before diving into specific aspects.
Specific comments:
L398-399: The text states that the control diet (C-Sal) is distributed along the positive side of LDA2. However, it appears that these observations are half on the negative side. Please review and correct if needed.
L487: S. aurata in italic
L546: ighm in italic
Author Response
Reviewer 2
The present study evaluated the effect of GHRP-6 as dietary additive in Spaurus aurata on endocrine and immune responses following IFA treatment. The manuscript sounds well-designed and written. It presents the background, aims, M&M, and results in a well-structured manner. The topic addressed by the authors is relevant, mainly in the context of modern aquaculture, which faces an emerging demand for more sustainable diets, and alternative additives to enhance fish health and resilience, avoiding the use of antibiotics, for instance.
The results suggest that GHRP-6 may affect positively both endocrine and immune responses, as demonstrated by changes in plasma biochemistry and gene expression. Specifically, following IFA treatment, fish fed the GHRP-6 supplemented diet showed greater resistance to stress, as evidenced by the stability of the stress-related biomarkers. In addition, there was an upregulation of genes associated with anti-inflammatory and immune responses. These and previous results may contribute to the development of more sustainable nutritional alternatives in aquafeeds.
The manuscript is well-written overall, and the figures and tables are informative and appropriately presented. Therefore, I suggest a few minor revisions:
Authors: Thank you for your thoughtful comments and valuable suggestions. We believe that your recommendations have significantly improved the overall quality and clarity of the manuscript. In response to your suggestions:
General concepts comments:
The introduction is well-structured; the reader can identify the problem underlying the objectives. The authors provide up-to-date and relevant literature supporting the use of GHRPs to improve growth, immune system, and defense responses in other teleosts. The objectives are well defined, and the rationale is well supported by the authors’ previous findings on feed efficiency and growth performance, supporting the new questions regarding its effect on endocrine and immune responses. The literature is current and consistent with the subject.
However, the authors should consider providing references for lines 58-59: “To solve this problem, generic antibiotics started to be introduced into animal feed quite a few years ago.”
Authors: Reference added
In the Material and Methods section, it would be important to provide to the readers some brief information regarding the plasma analyses, for example, the limit of the accepted coefficient of variation (CV%) between replicates, the wavelength used for readings, the inter and intra- assay variation, validation tests for the species, if available. The authors should as well include the references for the doses of both GHRP-6 (500 ug/kg of feed) and IFA (100 uL/100 g fish) used in the present study. This is particularly important for IFA, given that the authors later discuss that it could be a weak dose for triggering a stronger effect.
Authors: Information regarding the specific wavelengths used for measuring each parameter has been incorporated into the Materials and Methods section, following the guidelines provided in the protocols of each commercial kit used. Although these assays have not been specifically validated for Sparus aurata, they have been consistently applied in this species by our research group over several years, yielding reliable and reproducible results. The acceptable coefficient of variation (CV%) between replicates is also indicated.
The dose of GHRP-6 used in this study (500 µg/kg feed) was selected based on our previous work (Rodríguez-Viera et al., 2022; Martínez et al., 2017, 2016), which demonstrated positive effects on growth performance and immune-related parameters. Notably, both the present study and Rodríguez-Viera et al. (2022) represent the first reports of GHRP-6 being administered as a dietary supplement in fish.
The IFA dose (100 µL/100 g fish) was chosen based on previous immunostimulation studies in teleosts (e.g., Mutoloki et al., 2008; Gjessing et al., 2012). However, to the best of our knowledge, IFA has not previously been used in Sparus aurata. In fact, we are aware of only one study reporting its application in Atlantic cod. Therefore, our work provides a preliminary reference point for its potential use in S. aurata. We acknowledge that, under certain conditions, this dose may elicit a moderate immune response, as discussed in the manuscript.
The results and discussion are easy to follow, figures and tables are well presented. If possible, it would be valuable to include micrographs of each morphometric parameters in the intestine and spleen analyzed as a supplementary figure, to better illustrate the histological observations. Although the first paragraph of the discussion is very informative, I suggest briefly summarizing the main findings overall to position the reader and emphasize the study’s significance before diving into specific aspects.
Authors: Micrographs illustrating each morphometric parameter analyzed in the intestine and spleen have been included as supplementary figures to better support the histological observations. Additionally, the first paragraph of the Discussion section has been revised to include a brief summary of the main findings, providing better context and emphasizing the significance of the study before addressing specific aspects.
Specific comments:
L398-399: The text states that the control diet (C-Sal) is distributed along the positive side of LDA2. However, it appears that these observations are half on the negative side. Please review and correct if needed.
Authors: It was clarified in the text
L487: S. aurata in italic
Authors: Corrected.
L546: ighm in italic
Authors: Corrected.
Reviewer 3 Report
Comments and Suggestions for Authors
Specific comments:
- The introduction section is too long. I recommend that the authors consider condensing the introduction and focusing on the research's main topic.
- Line 129: Although you have used a commercial diet, please indicate the approximate composition of the diet, including the amounts of protein, fat, etc.
- Line 147-151: Did you use the anesthetics on a previously established protocol? If yes, please cite the appropriate reference.
- The Discussion section does not need such a big introductory paragraph, as this information is given in the manuscript's Introduction section.
- Lines 463-467: More references should be cited to support this hypothesis.
Author Response
Reviewer 3
- The introduction section is too long. I recommend that the authors consider condensing the introduction and focusing on the research's main topic.
Authors: Thank you very much for your constructive comments. The introduction was written in response to the three reviewers' suggestions, incorporating all their feedback. An attempt was made to condense it further, as suggested by the reviewer.
- Line 129: Although you have used a commercial diet, please indicate the approximate composition of the diet, including the amounts of protein, fat, etc.
Authors: Thank you for pointing this out. Information was added to the text.
- Line 147-151: Did you use the anesthetics on a previously established protocol? If yes, please cite the appropriate reference.
Authors: Yes, the anesthetic protocol using 2-phenoxyethanol (0.3 mL/L seawater) was applied following previously established protocols validated for Sparus aurata. Specifically, the methodology was based on Rodríguez-Viera et al. (2022) and adapted from methods described in Afonso et al. (2012), which demonstrated the effectiveness and safety of this anesthetic for short-term procedures in this species. These references have been added to the Materials and Methods section accordingly.
- The Discussion section does not need such a big introductory paragraph, as this information is given in the manuscript's Introduction section.
Authors: We agree with your observation. The long introductory paragraph of the discussion was removed.
- Lines 463-467: More references should be cited to support this hypothesis.
Authors: We agree with your observation, and additional references have been included to better support the hypothesis discussed.
Authors: Thank you for your thoughtful comments and valuable suggestions. We believe that your recommendations have significantly improved the overall quality and clarity of the manuscript.
Round 2
Reviewer 3 Report
Comments and Suggestions for Authors
Thank you for revising. The manuscript has been revised appropriately.